# Theoretical Study of the Motion of a Cut Sugar Beet Tops Particle along the Inner Surface of the Conveying and Unloading System of a Topping Machine

Simone Pascuzzi [1,*], Volodymyr Bulgakov [2], Ivan Holovach [2], Semjons Ivanovs [3], Aivars Aboltins [3], Yevhen Ihnatiev [4], Adolfs Rucins [3], Oleksandra Trokhaniak [2] and Francesco Paciolla [5]

1 Department of Soil, Plant and Food Science, University of Bari Aldo Moro, Via Amendola 165/A, 70126 Bari, Italy
2 Department of Mechanics, Faculty of Construction and Design, National University of Life and Environmental Sciences of Ukraine, 15 Heroiv Oborony Str., 03041 Kyiv, Ukraine; vbulgakov@meta.ua (V.B.); holovach.iv@gmail.com (I.H.); klendii_o@ukr.net (O.T.)
3 Institute of Engineering and Energetics, Faculty of Engineering and Information Technologies, Latvia University of Life Sciences and Technologies, Cakstes Blvd. 5, 3001 Jelgava, Latvia; semjons@apollo.lv (S.I.); aivars.aboltins@lbtu.lv (A.A.); adolfs.rucins@lbtu.lv (A.R.)
4 Department of Operation and Technical Service of Machines, Dmytro Motornyi Tavria State Agrotechnological University, 66 Zhukovsky Str., 69600 Zaporizhzhia, Ukraine; yevhen.ihnatiev@tsatu.edu.ua
5 Department of Electrical and Information Engineering (DEI), Polytechnic University of Bari, Via Orabona 4, 70126 Bari, Italy; francesco.paciolla@poliba.it
* Correspondence: simone.pascuzzi@uniba.it

**Abstract:** One of the most delicate operations in the sugar beet harvesting process is removing the tops from the heads of the root crops without any mechanical damages. The aim of this study is to improve the design of the conveying and unloading system of the sugar beet topper machine. In this paper, a mathematical model of the motion of a cut beet tops particle $M$, along the conveying and unloading system, has been developed to support the evaluation of kinematic and design parameters, depending on the rotational speed of the thrower blade, the air flow speed, the required ejection speed of particle $M$, and the position of the trailer that moves alongside the harvester. It has been established that increasing the speed $Va$ of the top particle M, which has left the end of the blade of the thrower, leads to an increase in the arc coordinate $S(t)$ of its movement along the cylindrical section of the casing. Within the range of the speed change from 4 m·s$^{-1}$ to 8 m·s$^{-1}$, the value of the arc coordinate $S(t)$ increases by 1.4 times during time $t$ = 0.006 s. Moreover, a rapid decrease in speed $V$ is observed with an increase in the length $x$ of the discharge chute.

**Keywords:** sugar beet; top removal; loading; mathematical simulation; rational parameters

## 1. Introduction

According to several reports, starting from 2013, more than one hundred countries around the world produce sugar, and about half of them produce it from sugar beet [1–4]. These countries, such as those in Western, Central, and Eastern Europe, the United States, China, and Japan, have moderate climate conditions. European Union countries, the Russian Federation, and the United States represent the biggest producers of sugar beet [1–4]. Global sugar beet production is projected to reach 284 Mt by 2031, with a slower annual growth rate of about 0.2% per year [5].

The sugar beet crop has two derivative products, beet pulp and molasses, which are employed for the manufacturing of different products in a wide range of applications [6]. Sugar beets are mainly used for sugar production in the food sector, but are also used to produce bio-based products for the pharmaceuticals, plastics, and textiles industries, and for ethanol production [7]. Sugar beet leaves and tops are used as high-quality animal feed

since they are rich in nutrients and are also used as raw material for biogas production [8]. Unfortunately, beet tops are often not collected after cutting but simply crushed and spread on the soil, making them unusable for animal fodder [9]. Furthermore, sugar beet leaves are also considered as an alternative protein source for the food industry; indeed, they are currently being considered as a novel food by the European food safety authority (NF 2021/2370) [7].

Due to the fragility of sugar beets and the stringent requirements for the quality of the final product, harvesting is a crucial operation [8–10]. Efficient and high-quality harvesting, using specialized machines such as multi-row self-propelled harvesters, is mandatory [11–14]. Nowadays, some modern harvesters have the possibility of installing adaptive devices based on computer vision systems, by which a significant amount of losses during the harvesting process can be reduced, thanks to, for instance, a digital two-dimensional recording system combined with a convolutional neural network (CNN) for detecting defects in harvested sugar beets [15,16].

To increase the efficiency and productivity of the harvesting process, it is necessary to develop new machines for precise cutting of the tops, considering that sugar beet heads have different sizes, shapes, and heights from the soil surface [17].

It is important to highlight the importance of adequately preparing the production surface to limit the influence of soil roughness, which causes oscillations in the vertical plane of front-mounted beet topper machines, ensuring the precision of the cutting. Experimental tests point out that cutting uniformity and correct harvesting without mechanical damage and losses are strongly affected by machine and cutting device oscillations, mainly caused by soil unevenness, position of feeler wheels, and tractor speed [18].

The sugar beet harvesting process presents several problems, with one of the most urgent being removing the tops from the heads of the root crops and their transportation without any mechanical damage and losses [19–21].

To solve this issue, it is necessary to develop new advanced designs for the loading mechanism used to collect the cut sugar beet tops and unload them through the discharge chute into a vehicle that moves sideways to the harvester. To perform a rational design and an exact evaluation of the kinematic parameters of the conveying and unloading system, it is necessary to conduct deep theoretical and experimental studies of the cut sugar beet tops loading process.

The particles of cut sugar beet tops, unlike, for example, particles of chopped herbs and silage corn, have quite different shapes, sizes, weights, humidity levels (parts of the tops are already dry during harvesting), densities, and friction coefficients. Furthermore, naturally there is sugar juice, which "glues" the sugar beet particles together during the conveying and unloading process.

Due to these circumstances, few researchers have analytically examined the motion of these particles inside the inner surface of the conveying and unloading system of a top harvesting machine.

Fundamental theoretical studies of the motion of cut sugar beet tops from the cutting mechanism to the blade thrower, and along the blade of the thrower into the discharge chute, have been reported in [22,23].

The study of the motion of a cut sugar beet tops particle (hereafter, particle *M*) after it has left the thrower blade and before leaving the discharge chute has been carried out in this paper. A mathematical model, which includes all the design parameters of the conveying and unloading system and the forces acting on particle *M*, which moves along the inner surface of the aforesaid system, has been developed. This model allows us to obtain the exact kinematic parameters of a particle *M*, which are also useful to understand its further motion outside the harvesting machine.

It should be remarked that some theoretical studies of the motion of waste material particles along the surfaces of the working bodies of agricultural machines are reported in previous works [24–28].

Other papers [29–32] have reported on the influence of the design parameters of the tops harvesting machine on the quality of the harvesting operation. However, the motion of particle *M* along the casing of the unloading mechanism is not considered at all.

The issues related to the motion and sliding of particle *M* along the surface of the working parts of agricultural machines are partly considered in works [33–36], but no attention is paid to the influence of the external environment in which the particle is moving.

Some other theoretical studies about the resistance exerted by environmental conditions on the motion of particle *M* have also been reported in [37,38]. Nevertheless, it is not possible to use them for analyzing and simulating the motion of particle *M* due to the complex shape of the unloading mechanism.

Finally, previous studies of particle *M* motion do not provide sufficient accuracy because they do not take into account all the design parameters of the beet tops harvester and the influence of environmental conditions [24–26], even if these are fundamental for the correct design of the unloading mechanism.

Taking in mind the aforesaid, this paper develops a mathematical model of the motion of particle *M* along the inner surface of the conveying and unloading system of a beet tops harvesting machine, considering the influence of the design parameters of the system, the kinematic parameters of particle *M*, and the influence of the airflow. In the following Materials and Methods section, the mathematical model of the motion of a cut beet tops particle *M* along the conveying and unloading system is defined. In the Results and Discussion sections, the graphical dependencies obtained by solving the differential equations of the motion of particle *M* are presented and analyzed.

## 2. Materials and Methods

The theoretical study presented in this paper is based on theoretical mechanics methods [39,40], which have allowed for the development of mathematical models that have been implemented in an algorithm, thanks to which mathematical simulations and graphical dependences have been obtained.

### 2.1. The New Front-Mounted Tractor Beet Topper Machine

We have developed a new front-mounted tractor beet topper machine (Figure 1a), which chops the leaves placed on the sugar beet tops in a never-ending way, conveying and unloading the cut sugar beet tops into a trailer that follows sideways. The conveying system essentially consists of a thrower blade (Figure 1b). The thrower blade (Figure 1b—6) splits up the leaves placed on them, conveying them to the unloading system, which consists of a cylindrical section (Figure 1b—2) and a discharge chute (Figure 1b—3), through which they are unloaded into a trailer that moves beside the sugar beet tops harvester.

The developed mathematical model considers the motion of particle *M* after it has left the thrower blade, produced by the force of the airflow generated by the rotation of the blade thrower, along the inner surface of the conveying and unloading system. This system is composed of a cylindrical and a rectangular straight-line section represented by the discharge chute.

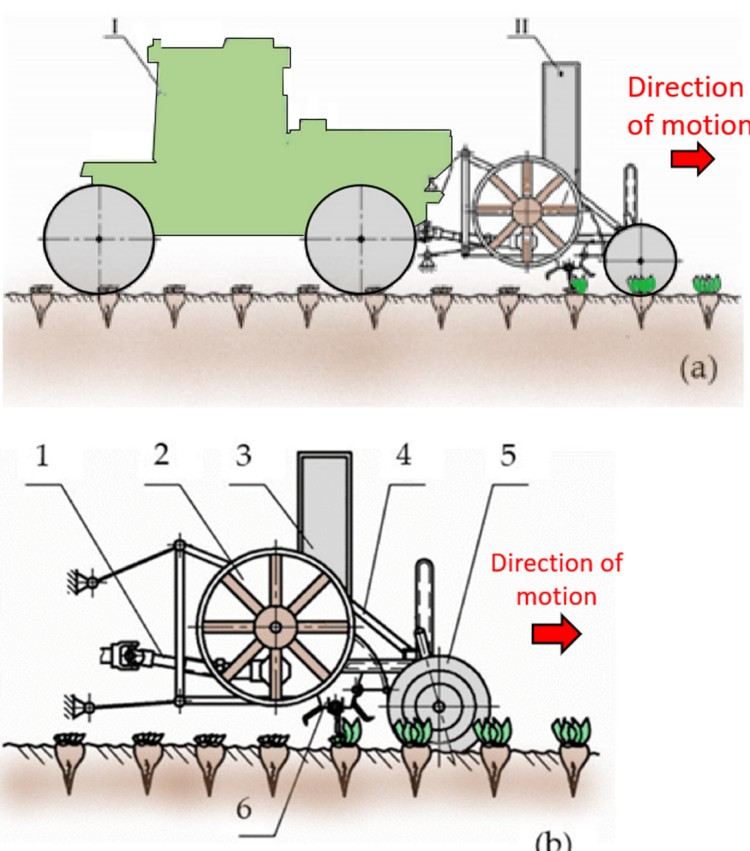

**Figure 1.** Front-mounted beet topper machine: (**a**) overall view: I—tractor; II—beet topper implement; (**b**) machine main scheme: 1—drive working bodies; 2—cylindrical section of the unloading system; 3—discharge chute; 4—frame; 5—pneumatic feeler wheel; 6—rotary cutting device.

*2.2. Mathematical Model of Motion of Particle M along the Cylindrical Section of the Conveying and Unloading System*

Firstly, a differential equation for the motion of particle *M* along the cylindrical section of the system after it has left the blade of the thrower was defined. To achieve this scope, a force diagram concerning the motion of particle *M* in the cylindrical section of the system was developed and is depicted in Figure 2.

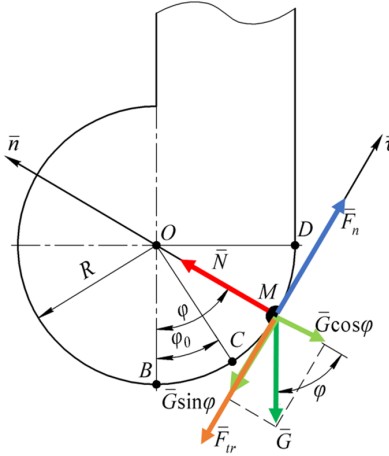

**Figure 2.** Force diagram concerning the motion of particle *M* after it has left the blade of the thrower in the cylindrical section of the conveying and unloading system. Aerodynamic force $\overline{F}_n$ (blue line), weight force $\overline{G}$ (green line), normal reaction force $\overline{N}_1$ (red line), friction force $\overline{F}_{tr1}$ (orange line).

It is convenient to describe the position of particle *M* at an arbitrary time *t* on the cylindrical section of the conveying and unloading system through the angular coordinate $\varphi$, as shown in the force diagram (Figure 2). Moreover, it is convenient to start the measurement of the angle $\varphi$ from the vertical ($\varphi = 0$ at point *B*), considering positive angles in the direction of the motion of particle *M*. Therefore, at time $t_0$, when particle *M* leaves the thrower blade (point *C* in Figure 2), the angle is defined as $\varphi_0$. Furthermore, *S* is the arc length covered by particle *M* while it is moving along the cylindrical section *CD* (Figure 2). When particle *M* exits from the cylindrical section (point D in Figure 2), the angle $\varphi$ is equal to $\varphi = \pi \cdot 2^{-1}$. The generic arc length covered by particle *M* is equal to:

$$S = \varphi \cdot R. \tag{1}$$

To describe the motion of particle *M* in section *CD*, it is possible to define a local coordinates system $M\tau n$, in which $\tau$ is tangent to the curve *CD* at point *M*; *n* is the normal to the curve *CD* at point *M*, and *CD* is an arch of the cylindrical section (Figure 2).

The force diagram concerning the forces that act on particle *M* during its motion along the cylindrical section *CD* is composed as follows:

○ The weight force $\overline{G}$ of particle *M*, whose magnitude is given by:

$$\overline{G} = m \cdot \overline{g}. \tag{2}$$

○ The normal reaction $\overline{N}$ of the inner surface of the cylindrical section, whose magnitude will be determined later.
○ The sliding friction force $\overline{F}_{tr}$ of particle *M* along the inner surface of the cylindrical section, whose magnitude is given by:

$$\overline{F}_{tr} = f \cdot \overline{N}. \tag{3}$$

○ The aerodynamic force $\overline{F}_n$ of the airflow, generated by the rotating blade of the thrower, which works as a fan. This force is determined according to the following equation [20]:

$$\overline{F}_n = k \cdot \left(\overline{V}_n - \overline{V}\right)^2. \tag{4}$$

According to [21], the air coefficient *k* is given by:

$$k = c_d \cdot \rho_a \cdot A_{cs}. \tag{5}$$

By substituting (5) into (4), it gives:

$$\overline{F}_n = c_d \cdot \rho_a \cdot A_{cs} \cdot \left(\overline{V}_n - \overline{V}\right)^2. \tag{6}$$

Considering the system of forces presented in the force diagram, the equation of motion of a cut beet tops particle *M* in a vector form is given as follows:

$$m\overline{a} = \overline{G} + \overline{N} + \overline{F}_{tr} + \overline{F}_n. \tag{7}$$

where $\overline{a}$ is the acceleration associated with the motion of particle *M*.

Projecting Equation (7) onto axes $M_\tau$ and $M_n$, respectively, of the local coordinate system $M\tau n$, and considering the intensity of each force, yields the following system of differential equations:

$$\left. \begin{array}{l} m \cdot \ddot{S} = -G \cdot \sin \varphi - F_{tr} + F_n \\ \frac{m \cdot \dot{S}^2}{R} = N - G \cdot \cos \varphi \end{array} \right\} \tag{8}$$

From the second equation of system (8), the normal reaction $N$ is given by:

$$N = m \cdot g \cdot \cos \varphi + \frac{m \cdot \dot{S}^2}{R}. \tag{9}$$

The friction force $F_{tr}$, considering Equations (3) and (9), is given by:

$$F_{tr} = f \cdot N = f \cdot \left( m \cdot g \cdot \cos \varphi + \frac{m \cdot \dot{S}^2}{R} \right). \tag{10}$$

Considering Equations (4) and (10), the first equation of the system (8) is given by:

$$m \cdot \ddot{S} = -m \cdot g \cdot \sin \varphi - f \cdot \left( m \cdot g \cdot \cos \varphi + \frac{m \cdot \dot{S}^2}{R} \right) + k \cdot \left( V_n - \dot{S} \right)^2. \tag{11}$$

Dividing Equation (11) by the mass $m$ and carrying out a series of mathematical transformations, the following equation is given:

$$\ddot{S} + \frac{2k \cdot V_n}{m} \cdot \dot{S} + \left( \frac{f}{R} - \frac{k}{m} \right) \cdot \dot{S}^2 = -g \cdot (\sin \varphi + f \cdot \cos \varphi) + \frac{k}{m} \cdot V_n^2. \tag{12}$$

Considering Equation (1), Equation (12) can be rearranged as follows:

$$\ddot{S} + \frac{2k \cdot V_n}{m} \cdot \dot{S} + \left( \frac{f}{R} - \frac{k}{m} \right) \cdot \dot{S}^2 = -g \cdot \left[ \sin \left( \frac{S}{R} \right) + f \cdot \cos \left( \frac{S}{R} \right) \right] + \frac{k}{m} \cdot V_n^2. \tag{13}$$

The initial conditions imposed at $t = 0$ to solve the differential Equation (13) are the following:

$$\dot{S}(0) = V_a \text{ and } S(0) = S_0. \tag{14}$$

Equation (13) has been solved on the arc $CD$ of the cylindrical section, from point $C$ to point $D$, which represents the end of the cylindrical section and the beginning of the discharge chute. Since Equation (13) does not include the angular coordinate $\varphi$ but the arc length $S$, the solution of Equation (13) has been performed up to the value $S_1$, thus up to point $D$.

The differential Equation (13) is nonlinear and can only be solved by numerical methods. As a result, the graphical dependencies $S = S(t)$ and $\dot{S} = \dot{S}(t)$ have been obtained. The time $t_1$ represents the instant at which particle $M$ passes from the cylindrical section to the discharge chute of the conveying and unloading system, and the corresponding values of $S$ and $\dot{S}$ are $S = S_1$ and $\dot{S} = \dot{S}_1$, respectively. The initial speed of particle $M$ in the discharge chute is $V_1 = \dot{S}_1(t_1)$.

### 2.3. Mathematical Model of Motion of Particle M along the Discharge Chute

The next step is to write the differential equation that describes the motion of particle $M$ along the straight-line section $DL$ of the discharge chute.

For this purpose, a new force diagram (Figure 3) has been defined to study the forces that act on particle $M$ in section $DL$.

To describe the motion of particle $M$ in this section, a one-dimensional local coordinate system $Dx$ has been fixed, in which the $x$-axis is directed vertically upward, i.e., in the direction of motion along the discharge chute. The position of particle $M$ at an arbitrary time $t$ is described by this local coordinate system.

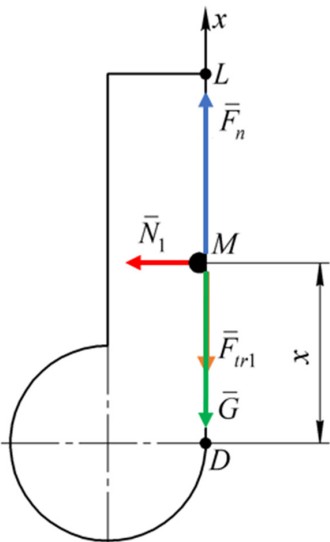

**Figure 3.** Force diagram concerning the motion of particle *M* along the straight-line section *DL* of the discharge chute. Aerodynamic force $\overline{F}_n$ (blue line), weight force $\overline{G}$ (green line), normal reaction force $\overline{N}_1$ (red line), friction force $\overline{F}_{tr1}$ (orange line).

The surface *DL* of this straight-line section is placed tangentially to the cylindrical section at point *D*.

Particle *M* is subjected to the weight force $\overline{G}$, determined according to Equation (2) and to the aerodynamic force $\overline{F}_n$, determined according to Equation (4).

As particle *M* moves along the inner side of the cylindrical section due to its inertial properties, in the same way, it moves upward along the inner surface *DL* of the discharge chute, or, in an extreme case, at a very close distance from it without adhering to it. The aerodynamic force presses particle *M* against the side of the inner surface *DL*. As a result, during its motion along the discharge chute, particle *M* is also subjected to the normal reaction force $\overline{N}_1$, arising from the side of surface *DL*, and to the friction force $\overline{F}_{tr1}$ (Figure 3).

Referring to this, the motion of particle *M*, pressed against the wall of a vertical discharge chute with a rectangular cross-section, has been considered. In a real tops harvester, the entire discharge chute is filled with a mass of mown tops that move upward and press against the inner side of the unloading discharge chute. Ideally, if there was only one particle, it would move strictly vertically upward and not come into contact with any side of the discharge chute. In real conditions, inside the discharge chute, streams of particles *M* move upward, constantly pressed against the sides.

At a first approximation, the normal reaction $\overline{N}_1$ has been considered constant throughout the entire section, from point *D* to point *L*, since cut particle *M*'s motion occurs over a fairly short period of time.

Consequently, the friction force is given by:

$$\overline{F}_{tr1} = f \cdot \overline{N}_1 = \text{const.} \tag{15}$$

Considering the system of forces presented in the force diagram (Figure 3), the equation in vector form of the motion of particle *M* in the straight-line section *DL* of the discharge chute is given by:

$$m \cdot \overline{a} = \overline{G} + \overline{F}_n + \overline{F}_{tr1} \tag{16}$$

Projecting Equation (16) onto the axis *Dx*, and considering Equations (2), (4), and (15) and the intensity of each force, gives:

$$m \cdot \ddot{x} = -m \cdot g + k \cdot (V_n - V)^2 - f \cdot N_1 \tag{17}$$

where $V = \dot{x}$, Equation (17) is rearranged in the following form:

$$m \cdot \ddot{x} = -m \cdot g + k \cdot (V_n - \dot{x})^2 - f \cdot N_1 \tag{18}$$

After some transformations, Equation (18) gives the following:

$$\ddot{x} + \frac{2k \cdot V_n}{m} \cdot \dot{x} - \frac{k}{m} \cdot \dot{x}^2 = \frac{k}{m} \cdot V_n^2 - g - \frac{f}{m} \cdot N_1 \tag{19}$$

Differential Equation (19) is nonlinear and can be solved only by numerical methods using a PC.

As a solution of the differential Equation (19), graphical dependencies $x = x(t)$ and $\dot{x} = \dot{x}(t)$, have been obtained. It is assumed that at time instant $t_2$, particle $M$ is ejected by the discharge chute. At time instant $t_2$, $x = x_2$ and $\dot{x} = \dot{x}_2 = V_2$. The speed $V_2$ is the initial speed of flight of tops particle $M$ from the discharge chute.

## 3. Results

In the numerical resolution, the motion of particle $M$ in the conveying and unloading system, starting from the instant at which the particle has left the blade of the thrower, has been studied.

Different graphs have been obtained by solving the differential Equations (13) and (19), considering the radius $R$ of the cylindrical section of the conveying and unloading system equal to 0.35 m. Furthermore, the air flow speed, created by the blade of the thrower, has been established as constant and equal to 5 m·s$^{-1}$.

### 3.1. Cylindrical Section of the Conveying and Unloading System

In the cylindrical section of the conveying and unloading system, particle $M$ moves only within a small angle range, from 0 rad (point $B$ in Figure 2) to 1.2 rad (point $M$ in Figure 2). Considering the cylindrical section radius and the angle range, the length of the arc path traveled by particle $M$ is 0.42 m. Thus, considering the real working rotational velocity of the blade equal to 40 rad·s$^{-1}$, the time spent by the particle on the inner surface of the cylindrical section is approximately 0.0065 s [40].

Referring to the cylindrical section of the conveying and unloading system and considering the aforesaid rotational speed of the blade, the graph presented in Figure 4 points out the dependence of the arc coordinate $S(t)$ on time $t$, during the motion of particle $M$ after it has left the thrower blade, considering different speeds $V_a$. The red dashed line in Figure 4 represents the actual arc length performed by particle $M$ to achieve the end of the cylindrical section.

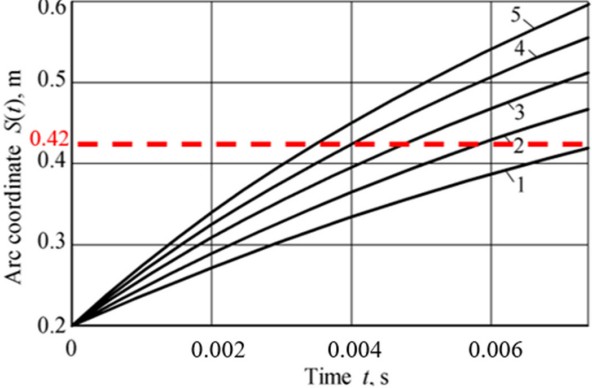

**Figure 4.** Dependence of the arc coordinate *S(t)* of particle *M* on time *t*, during its motion along the cylindrical section of the conveying and unloading system after it has left the thrower blade at the following speeds $V_a$: 1—$V_a = 4$ m·s$^{-1}$; 2—$V_a = 5$ m·s$^{-1}$; 3—$V_a = 6$ m·s$^{-1}$; 4—$V_a = 7$ m·s$^{-1}$; 5—$V_a = 8$ m·s$^{-1}$.

### 3.2. Straight-Line Section of the Conveying and Unloading System

The graph in Figure 5 reports the dependence of the speed $V$ of particle $M$ on time $t$ along the straight-line section $DL$ of the discharge chute, considering an initial speed $V_1 = 7$ m·s$^{-1}$. This speed value was measured during field tests conducted on the inner surface of the conveying and unloading system of a topping machine (Figure 1), considering the actual working rotational velocity of the blade as equal to 40 rad·s$^{-1}$ [40].

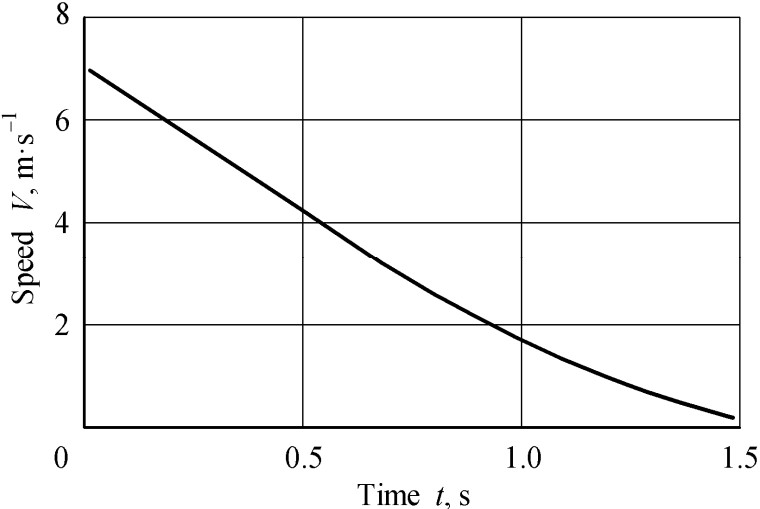

**Figure 5.** Dependence of the speed $V$ of particle $M$ on time $t$ along the straight-line section $DL$ of the discharge chute at a given initial speed of $V_1 = 7$ m·s$^{-1}$.

The time scales on the $x$-axis of Figure 4 (from 0 to 0.0065 s) and Figure 5 (from 0 to 1.5 s) are not comparable due to the different lengths of the cylindrical section and the discharge chute, respectively.

The dependence of the speed $V$ of particle $M$ on the length $x$ of the straight-line section DL of the discharge chute has been reported in Figure 6, considering an initial speed of $V_1 = 7$ m·s$^{-1}$.

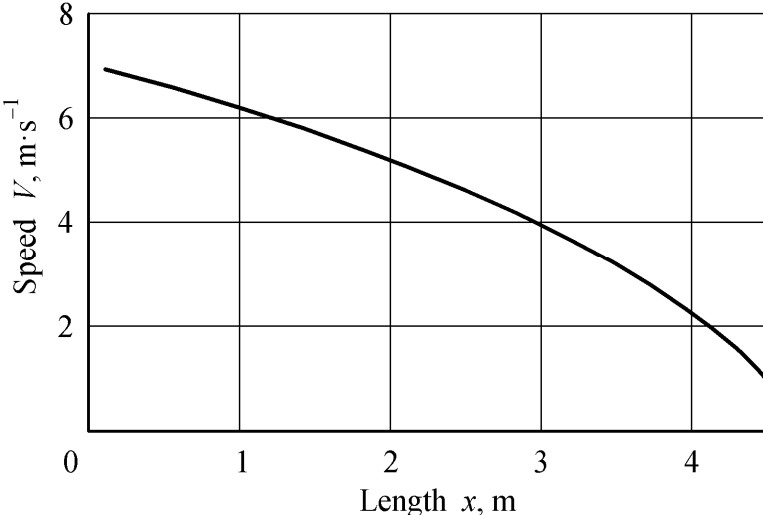

**Figure 6.** Dependence of the speed $V$ of particle $M$ on the length $x$ of the straight-line section $DL$ of the discharge chute, with an initial speed of $V_1 = 7$ m·s$^{-1}$.

### 4. Discussion

In the developed conveying and unloading system employed for trailed sugar beet tops harvesters, some wet and sticky particles $M$, due to the presence of sugar juice, can

accumulate inside the casing in which the four-blade thrower is located. Sometimes, this can lead to a decrease in the relative speed of motion of particle *M* and, consequently, to a decrease or a complete stop in the unloading process.

When unloading particles of various crops, such as green grass and corn, with similar mowing and loading machines, due to significantly less moisture values, this phenomenon is not observed. Thus, it is interesting to analytically study the conveying and unloading process of a sugar beet tops harvesting machine.

To select the best geometric and kinematic parameters of the conveying and unloading system, the differential Equations (13) and (19), which describe the motion of a cut sugar beet tops particle *M* along the cylindrical and straight-line sections of the conveying and unloading system of a topping machine, have been numerically solved. As a result, different graphical dependencies for the evaluation of kinematic parameters of particle *M* have been obtained.

The graphical dependence shown in Figure 4 highlights that an increase in the speed $V_a$ at which particle M has left the thrower blade leads to an increase in the arc coordinate S(t) traveled by particle M. Varying $V_a$ from 4 m·s$^{-1}$ to 8 m·s$^{-1}$, the respective arc coordinate traveled, after $t = 0.006$, increases almost by 1.4 times, passing from 0.39 m to 0.54 m.

As reported in Figures 4 and 5, the time of motion of particle *M* in the cylindrical section and in the straight-line section are not comparable because of the different lengths of the two sections. Thus, in the cylindrical section, the path traveled by particle *M* is very short, whereas the discharge chute is much longer and the speed drops longer. Furthermore, the graphical dependence in Figure 5 shows a non-linear decrease in the motion speed *V* of particle *M* over time *t*.

Regarding Figure 6, it is possible to conclude that, considering a discharge chute length of 4 m, the initial speed of flight $V_2$ of particle *M* from the discharge chute will be greater than 2 m·s$^{-1}$. If a greater length of the discharge chute is planned, it is necessary to increase the initial speed $V_1$ of particle *M* or provide a higher speed of the associated airflow.

Few studies report on the cutting, conveying, and unloading process of sugar beet tops. In [41], a multi-roll mechanism for leaf removal and a fixed-thickness cutting method have been studied. Moreover, the design of the leaf removal device has been optimized, and the defoliation scheme of the two roller shafts has been determined, considering the rotation speed of the roller shaft equal to 800 rpm and the forward speed of 0.8 m·s$^{-1}$. The mechanical analysis of the cutting device, with an inclined profile plate structure, has been analyzed, characterized by a profiling angle in the range of 25°–45° for the profiling plate and a cutting angle of 20°–40°. Compared with the 4TSQ-2 machine, the structural size of the machine has been reduced by 30%, the weight by 15%, the cost by 25%, the top cutting speed was increased by 2.6%, and the sugar beet growth has been reduced by 0.5%.

The study presented in [42] has been conducted at the Safiabad Agricultural Research Center of Dezful, Iran, to determine the effect of delayed loading of harvested beets on sugar yield loss. The highest loss (27.5%) has been registered for treatment 13, in which the leaves were first cut and removed, and then topdressing (removal of the crown) by uprooting the beets and leaving them in the field for 48 h before loading. In the control treatment, beets were loaded soon after harvesting. Sugar content in beets was the highest (18.71%) and the lowest (15.48%) in treatments 13 and 9 (similar to option 13, but seeding was delayed 24 h after harvest), respectively. The sugar yield was the highest in the control (12.37 t·ha$^{-1}$) and the lowest in 13 (10.54 t·ha$^{-1}$). The conclusions drawn were that any delay in loading and uprooting beet roots with tops could cause high yield losses in the Dezful area.

A comparative assessment of different agricultural machines that mow, transport, and unload plant materials is presented in Table 1.

**Table 1.** Specifications of machinery for collecting and trimming mown crops.

| Machinery | Speed | Performance | Advantages | Limitations |
|---|---|---|---|---|
| 1. Forage harvesters (trailed) [43] | $2.3 \div 3.3$ m·s$^{-1}$ | $50 \div 90 \cdot 10^3$ kg·h$^{-1}$ | The slatted system ensures the unloading of corn in any condition. | Design complexity, high energy consumption. |
| 2. Hay mowers (trailer) [44] | up to 3.0 m·s$^{-1}$ | $2.5 \div 4.5$ ha·h$^{-1}$ | Cut and unload in a windrow on the ground. | Additional technical and operating costs for further collection or pressing of the roll |
| 3. Mower-choppers (trailer) [45] | up to 2.2 m·s$^{-1}$ | $15 \div 18 \cdot 10^3$ kg·h$^{-1}$ | Cut and unload in a windrow the remains of sunflower, sorghum, potato tops, corn, and grass. | Difficult to operate, since the unit has a trailer connected to the rear to collect the mown mass. |
| 4. Topper removal machines (self-propelled) [46] | $1.8 \div 2.5$ m·s$^{-1}$ | $1.0 \div 1.3$ ha·h$^{-1}$ | No conditions for the unloading system clogging at any moisture content. | Design complexity, high energy consumption due to the presence of a slatted unloading system. |
| 5. Design presented in the article | $2.5 \div 3.0$ m·s$^{-1}$ | up to 2.0 ha·h$^{-1}$ (considering a working width of 1.35 m, three rows of sugar beet crops) | Simplicity of construction, and operational reliability. | A decrease or a complete stop in unloading process may occur because of accumulation of wet mass of tops which can clog the thrower. |

From the data presented in Table 1, it is possible to point out that the new tops harvesting machine, developed and proposed by the authors, has technical and operational indicators that are comparable with the best industrial machines. The theoretical research carried out has been fundamental for the correct evaluation of the design parameters of the machine. The developed design can be recommended for industrial production thanks to its acceptable performance indicators, simplicity of construction, and operational reliability.

## 5. Conclusions

The sugar beet harvesting process has several problems because of the fragility of the sugar beet tops, which can cause losses in sugar yield. Thus, one of the most important activities is removing the tops from the heads of the root crops and transportating them without any mechanical damage and losses.

The particles of cut sugar beet tops are wet and sticky due to the sugar juice, which "glues" the sugar beet particles together during the conveying and unloading process. Few researchers have deeply studied the motion of these particles inside the inner surface of the conveying and unloading system of a tops harvesting machine.

The aim of this study is to improve the design of the conveying and unloading system to find the most suitable architecture that allows for the best performance both in terms of efficiency of the machine and reduction of mechanical damage to the cut sugar beet tops. A mathematical model, represented by the two non-linear differential Equations (13) and (19), describing the motion of cut sugar beet tops particle *M*, starting from the working surface of the blades, along the conveying and unloading system, and finally during its free flight, has been developed. The developed model makes it possible to support the evaluation of kinematic and design parameters of the conveying and unloading system of the sugar beet topper machine, depending on the rotational speed of the thrower blade, the air flow speed, the required ejection speed of the sugar beet tops, and the position and the forward speed of the trailer that moves alongside the harvester.

As a result of numerical calculations, graphical dependencies for the motion of particle *M* along the cylindrical section and the discharge chute of the conveying and unloading system, have been obtained. In this research, the oscillations of the beet topper machine

during its motion, caused by the unevenness of the soil surface, have not been considered, but further analytical and experimental studies to better understand this aspect are being carried out.

**Author Contributions:** Conceptualization, V.B. and S.P.; methodology, S.P., V.B. and I.H.; software, A.A. and F.P; validation, V.B., I.H., O.T., S.I., F.P. and S.P.; formal analysis, V.B., O.T. and A.R.; writing—original draft preparation V.B., F.P, S.P. and Y.I.; writing—review and editing, V.B., S.I., A.A., Y.I. and F.P.; supervision, V.B. and S.P. All authors have read and agreed to the published version of the manuscript.

**Funding:** This research received no external funding.

**Data Availability Statement:** No new data were created or analyzed in this study. Data sharing is not applicable to this article.

**Conflicts of Interest:** The authors declare no conflicts of interest.

## Nomenclature

| | |
|---|---|
| $M$ | cut sugar beet top particle |
| $\varphi$ | angular coordinate |
| $B$ | initial position of $M$ in the inner surface of the cylindrical section |
| $C$ | point of the cylindrical section when $M$ leaves the thrower blade |
| $D$ | final position of $M$ in the inner surface of the cylindrical section and initial position of $M$ in the inner surface of the straight-line section (discharge chute) |
| $R$ | radius of the cylindrical section |
| $S$ | arc length described by $M$ during its motion along the cylindrical section |
| $t_0$ | time when $M$ is in $C$ |
| $\varphi_0$ | angular coordinate at $t_0$ |
| $S_0 = \varphi_0 \cdot R$ | arc length when $M$ is in $C$ |
| $S_1 = R \cdot \pi \cdot 2^{-1}$ | arc length when $M$ is in $D$ |
| $M\tau n$ | local coordinate system |
| $m$ | mass of the cut beet top particle |
| $\overline{g}$ | gravity acceleration |
| $f$ | friction coefficient |
| $\overline{V}_n$ | speed of air flow |
| $\overline{V}$ | speed of $M$ |
| $V_a$ | absolute speed at which the cut beet tops particle $M$ has left the thrower blade |
| $V_1$ | absolute speed at which the cut beet tops particle $M$ enters the discharge chute |
| $V_2$ | absolute speed at which the cut beet tops particle $M$ leaves the discharge chute |
| $k$ | coefficient affected by the physical and mechanical properties of $M$ |
| $c_d$ | dimensionless coefficient affected by the shape and the cross-sectional area of $M$ |
| $A_{cs}$ | cross-sectional area of $M$ |
| $\rho_a$ | air density |
| $\overline{a}$ | motion acceleration of $M$ |
| $\ddot{S}$ | tangential acceleration of $M$ (vector intensity) |
| $\dot{S} = V$ | speed of $M$ (vector intensity) |
| $\dfrac{\dot{S}^2}{R}$ | normal acceleration of $M$ (vector intensity) |

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
