# Peer review of "Theoretical Study of the Motion of a Cut Sugar Beet Tops Particle along the Inner Surface of the Conveying and Unloading System of a Topping Machine"

_agriengineering, doi:10.3390/agriengineering6010025_

Round 1

Reviewer 1 Report

Comments and Suggestions for Authors

Dear authors
For the sustainability of the magazine's high level of quality, I polite ask authors to put all sketches/diagrams/figures used in the paper in color (not black and white).

Best regards,

Author Response

Dear Reviewer,

in attach I send you a file.

Thank you veru much

Best regards

Simone Pascuzzi

Reviewer 2 Report

Comments and Suggestions for Authors

Dear authors,

First of all, I hope that you and your family are all healthy and well.

It is a great pleasure to review your interesting manuscript and contribute to its publication.

My analysis is organized as follows. Firstly, I present my main impression. Below I recommend the main comments.

I hope you find this review helpful in improving your manuscript.

_________________________________________

Main Impression

Thank you for your patience while I revised this manuscript. I appreciate the topic and find it relevant to develop an I the mathematical model of the motion of a cut beet tops particle M, along with the conveying and unloading system of a beet tops harvesting machine. I reviewed your manuscript to assess its quality and suitability for this journal. I appreciate the topic and the research effort you put into this research. Overall, the article has a coherent structure.

I recommend that authors improve the beginning of the abstract. The problem that will be investigated remains to be clarified.

At the end of the introduction, I recommend that authors add what will be covered in the next sections of the manuscript.

I recommend that authors split the "3. Results and Discussion" section.

I recommend that authors develop a discussion section in the manuscript. This needs to be done to discuss the evidence identified in this research in relation to other studies in the area.

I recommend that the authors add a table with the advantages and disadvantages of this system in relation to other cases identified in the literature - this will improve the originality and quality of the paper.

I recommend that the authors improve the conclusion of the manuscript. It remains to be added what are the limitations of this research.

Comments on the Quality of English Language

Minor editing of English language required.

Author Response

Dear Reviewer,

in attach I send you a file.

Best regards

Simone Pascuzzi

Round 2

Reviewer 2 Report

Comments and Suggestions for Authors

Hello dear authors,

I appreciate the effort you made in this review. Many recommendations I had were adjusted in this new version of the manuscript. However, I am still not satisfied with the discussion section of this paper. I recommend that authors relate the evidence from this work to existing studies in the area. This comparison between research and researchers needs to appear in this section. Furthermore, I recommend adding a table on the advantages and disadvantages of this system in relation to what already exists in the literature.

Comments on the Quality of English Language

A small English language edit is needed in the "discussion" section.

Author Response

Dear Reviewer,

in attached I send the file with our answers to your suggestions.

Thank you very much

best regards

Simone Pascuzzi

Round 3

Reviewer 2 Report

Comments and Suggestions for Authors

The authors followed all my recommendations, and the quality of the manuscript improved significantly. Congratulations on the research carried out by all of you.